# On sparse connectivity, adversarial robustness, and a novel model of the artificial neuron

**Abstract.** In this paper, we propose two closely connected methods to improve computational efficiency and stability against adversarial perturbations on contour recognition tasks: (a) a novel model of an artificial neuron, a "strong neuron," with inherent robustness against adversarial perturbations and (b) a novel constructive training algorithm that generates sparse networks with $O(1)$ connections per neuron.

We achieved an impressive 10x reduction (compared with other sparsification approaches; 100x when compared with dense networks) in operations count. State-of-the-art stability against adversarial perturbations was achieved without any counteradversarial measures, relying on the robustness of strong neurons alone.

Our network extensively uses unsupervised feature detection, with more than 95% of operations being performed in its unsupervised parts. Less than 10.000 supervised FLOPs per class is required to recognize a contour (digit or traffic sign), which allows us to arrive to the conclusion that contour recognition is much simpler that was previously thought.

**Keywords:** Sparse neural networks, unsupervised training, adversarial robustness

## 1 Introduction

In recent years, artificial neural networks have achieved impressive results on all computer vision benchmarks. Interestingly, this progress was made using two ideas that are many decades old: (1) an artificial neuron with a linear summator at its core and (2) stochastic gradient descent (SGD) training.

The combination of these ideas was fortuitous, allowing us to fit any decision function, no matter how complex. As a result, neural models surpassed human-level accuracy. However, we believe (and will justify below) that the very properties of summators and SGD impede progress in improving two other important metrics: the sparsity of the neural connections and adversarial stability.

In our work, we propose (1) a novel model of an artificial neuron with inherent robustness against adversarial perturbations and (2) a novel training algorithm that allows us to build extremely sparse networks with $O(1)$ connections per neuron. With these proposals, we achieved state-of-the-art performance and adversarial stability on a number of contour recognition benchmarks.

## 2    Related Work

Our work touches on several topics: unsupervised feature detection, network sparsification, and adversarial robustness.

Many approaches to unsupervised feature detection have been proposed. Our work follows [4] (learning convolution filters by clustering image patches with k-means). Other notable approaches include autoencoders [1], variational autoencoders [14], noise-as-target [2], learning features invariant under particular transformations [5], and local Hebbian learning [9].

The most popular sparsification strategy is pruning, either via $L_0/L_1$ penalization ([29], [10], [18]) or various explicit pruning strategies ([3], [12], [19]). Usually about 95-97% of weights are pruned ([27]).

Adversarial robustness is usually addressed via adversarial training ([20], [16], [25]). One notable approach is to use provable bounds on a network output under attack [15] for training. Another line of thought is to modify the basics of neural architecture in order to make it inherently robust (for instance, [17] proposes to use bounded ReLU).

## 3    The Novel Artificial Neuron ("Strong Neuron")

### 3.1    Contour Recognition = Logical AND + Logical OR

Contour recognition is an important subset of computer vision problems. It is deeply connected with properties of our world — we live in a universe full of localized objects with distinctive edges. Many important problems are contour based: handwritten digit recognition, traffic light detection, traffic sign recognition and number plate recognition.

There are also non-contour tasks, however — for example, ones that can only be solved by gathering information from many small cues scattered throughout an image (e.g., distinguishing a food store from an electronics store).

Contour recognition has interesting mathematical properties: (a) it naturally leads to $[0, 1]$-bounded activities; (b) contours are localized and independent from their surrounding (e.g., a crosswalk sign is a crosswalk sign, even in desert or rainforest); (c) complex contours can be decomposed into smaller parts, that are contours too.

Our insight is that contour recognition is essentially a combination of two basic operations on low-level features (see Figure 1):

- logical AND (detection), which decomposes high-level features as combinations of several low-level ones, placed at different locations. Say, digit "5" can be represented as AND($5_{TOP}$,$5_{BOTTOM}$)
- logical OR (generalization), which allows detectors to be activated by more diverse inputs. Say, "5" can be replaced by more general OR(5,5,5)

**Fig. 1.** Contour recognition: AND + OR

## 3.2   What Is Wrong With Linear Summators?

First, standalone summator-based neuron is, in some sense, weak. It does not perform detailed evaluation of its inputs — all it sees is just their weighted sum, a position relative to the separating hyperplane. This means that lack of activity in one channel (absence of some critical feature) can be masked by increased activities in other channels. We will need a group of neurons working together in order to make sure that, say, a "face neuron" cannot be activated by many repetitions of just one face part. Our point here is not about computational efficiency, but about the fact that we try to model human intuition about vision by using elements with counterintuitive properties.

Second, summator-based implementation of the AND/OR logic is very brittle, especially in high-dimensional spaces. The neuron can be set to an arbitrarily high value (or, alternatively, zeroed) by feeding it with many small activities in different channels [8].

## 3.3   Our Proposal

We propose to use $f() = \min(A, B, \dots)$ to implement AND-logic, to use $f() = \max(A, B, \dots)$ to implement OR-logic and to combine both kinds of logic in a novel summator-free artificial neuron — "strong neuron":

$$F_{strong} = \min\left(\max_i(w_{0,i}x_i), \max_i(w_{1,i}x_i), \max_i(w_{2,i}x_i)\right) \qquad (1)$$

The formula above describes a strong neuron with three receptive areas, usually located at different parts of the input image. Here $w_{k,i} \in \{0, 1\}$ are extremely sparse binary weights — just $O(1)$ connections per neuron are generated during training (about 5-10 in most cases). Inputs $x_i$ are either initial low-level features corresponding to small (4x4, 5x5 or 6x6) image patches or medium-level features (computed by the previous layer of strong neurons).

We call our artificial neuron "strong" because it has a much more complex decision boundary than the summator-based neuron. It is not prone to the failure modes described in the previous subsection: (1) nonlinearities introduced by the $min$ and $max$ elements naturally align with human intuitive understanding of the pattern recognition; (2) the neuron is robust with respect to adversarial attacks

**Summator-based neuron**                    **Strong neuron**

Fig. 2. A summator-based neuron and a strong neuron

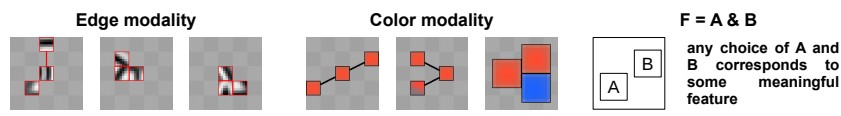

Fig. 3. Strong neurons are interpretable models

— an $\epsilon$-bounded perturbation of inputs produces exactly $\epsilon$-bounded perturbation of outputs.

Figure 3 visualizes several strong neurons generated during the solution of the German Traffic Sign Recognition Benchmark. One may see that strong neurons have obvious geometric interpretation. Actually, *any* (even completely random) strong neuron with spatially separate input areas corresponds to some meaningful feature. Some of these features are useful for prediction, most of them are useless — but all of them have obvious geometric meaning. This sharply contrasts with the properties of the convolutional filters — convolution with random coefficients rarely has clear geometric interpretation.

We would like to note here some connection with recent work on adversarial robustness, [28] and [13], which state that there are two kinds of features in images — highly predictive ones that are extremely unstable under unnatural perturbation of input, and less predictive but robust ones. The latter are the ones that are used by human vision; the former are imperceptible by humans but are heavily used by adversarially unstable neural networks. In particular, [28] shows an elegant example of both kinds of features derived from synthetic data. Viewed from this angle, strong neurons are constrained to a realm of geometric inference — and robust feature detectors.

## 4   The Motivation Behind Our Model

In this section, we will show that our artificial neuron model is motivated by some fundamental considerations, that is, there are some reasonable and intuitive requirements that are satisfied by our model — and are not satisfied by summator-based neurons.

First, we define the $L_\infty$-nonexpansive function as one which in a general N-dimensional case for any N-dimensional input perturbation $\Delta x$ satisfies

$$|f(x + \Delta x) - f(x)| \leq \max_i |\Delta x_i| = \|\Delta x\|_\infty \qquad (2)$$

The $L_\infty$-nonexpansive function has $\epsilon$-bounded perturbation of the output under $\epsilon$-bounded perturbation of its inputs, i.e. it does not accumulate perturbations (compare it with $L_1$-nonexpansivity that, despite "non" prefix, means that perturbations are summed up).

Human vision — and any artificial vision system that is intended to be robust — has a bounded reaction to bounded perturbations of the input image. The bounding ratio is not always 1:1 because sometimes we want to amplify weak signals. Thus, enforcing $L_\infty$-nonexpansivity on the entire classifier may overconstrain it. However, it makes sense to enforce this constraint at least for some parts of the classifier. One may easily show that both $min$ and $max$, as well as their superposition, are $L_\infty$-nonexpansive.

The rationale behind our model of the artificial neuron should be obvious by now — making inference as robust as possible. However, we present an even more interesting result — the fact that $min$ and $max$ are the only perfectly stable implementations of AND/OR logic.

One familiar with the history of artificial neural networks may remember the so-called "XOR problem" [21] — a problem of fitting the simple four-point dataset that cannot be separated by the single linear summator. Inspired by its minimalistic beauty, we formulate two similar problems, which address the accumulation of perturbations in multilayer networks:

*Theorem 1: $L_\infty$-nonexpansive AND problem.* $\exists!$ $f(x, y) = min(x, y)$ such that the following holds:

1. $f(x, y)$ is defined for $x, y \in [0, 1]$
2. $f(0, 0) = f(0, 1) = f(1, 0) = 0, \quad f(1, 1) = 1$
3. $a \leq A, \quad b \leq B \implies f(a, b) \leq f(A, B)$ (monotonicity)
4. $|f(a + \Delta a, b + \Delta b) - f(a, b)| \leq max(|\Delta a|, |\Delta b|)$

*Theorem 2: $L_\infty$-nonexpansive OR problem.* $\exists!$ $g(x, y) = max(x, y)$ such that the following holds:

1. $g(x, y)$ is defined for $x, y \in [0, 1]$
2. $g(0, 0) = 0, \quad g(0, 1) = g(1, 0) = g(1, 1) = 1$
3. $a \leq A, \quad b \leq B \implies g(a, b) \leq g(A, B)$ (monotonicity)
4. $|g(a + \Delta a, b + \Delta b) - g(a, b)| \leq max(|\Delta a|, |\Delta b|)$

Proofs of theorems 1 and 2 can be found in Appendix A (supplementary materials).

An immediate consequence of these theorems is that it is impossible to implement a robust AND (robust OR) element with just one ReLU neuron — the best that can be achieved is $L_1$-nonexpansivity, which is not robust. It is possible to 'emulate' robust AND/OR logic by performing tricks with many traditional ReLU neurons ($max(a, b) = a + ReLU(b - a)$, $max(a, b, c) = max(a, max(b, c))$ and so on), but the result will be just another implementation of $min$ and $max$ elements.

# 5   Contour Engine: Architecture Overview

The key parts of our neural architectures are outlined in Figure 4.

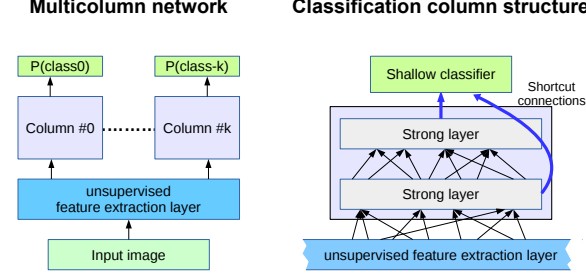

**Fig. 4.** The Contour Engine network

Strong neurons can perform logical inference on low-level features, but they cannot *produce* these features from raw pixel values. Thus, an initial feature extraction block is essential in order to "prime" the Contour Engine. This part of the network is the one that needs the most FLOPs — about 95% of the floating-point operations are spent in the unsupervised preprocessing.

The next part of our network is organized into many class-specific columns. Each column starts with sparse contour detection layers (one or two is usually enough) that combine low-level features in order to produce medium- and high-level features. Typical column widths are in $[50, 800]$ range. On average, less than 10.000 FLOPs per class are needed to recognize well-centered medium-complexity contour (like a digit, letter or traffic sign).

Finally, a shallow nonlinear classifier on top of each column post-processes the features produced by the robust contour detection stage. In practice, strong neurons are so good at contour detection that we do not need complex nonlinear models at this stage — the basic logistic model is enough.

The training algorithm includes three distinct, sequential stages:

- training unsupervised feature detector
- training sparse contour detection layers
- training a shallow classifier

We train a feature extraction layer using an unsupervised procedure (running k-means over image patches; see [4]). Such an approach makes the input layer independent from label assignment, which allows us to make some interesting conclusions regarding the asymptotic complexity of the image recognition.

Sparse layers are trained by adding layers and neurons one by one, in a greedy fashion, fitting new neurons to the current residual. The second important contribution of our work (in addition to the robust artificial neuron) is the heuristic,

which can efficiently fit nonsmooth strong neurons with binarity/sparsity constraints on weights.

Finally, the shallow classifier can be trained by running logistic regression over the activities of the sparse layers.

## 6    Training Feature Detection Layer

The purpose of our feature extraction layer is to describe the input image using a rich dictionary of visual words. The description includes features such as oriented edges, more complex shapes, colors and gradients, computed at multiple scales and orientations.

The key point of Coates et al. [4] is that one may achieve surprisingly good classification performance by processing images with a single convolutional layer whose filters are trained in an unsupervised manner (k-means on random image patches). Filters as large as 4x4, 5x5 or 6x6 typically give the best results.

We extend their results (see Figure 5) with:

- separate processing of color-agnostic (shape-sensitive) and color-based features by enforcing constraints on filter coefficients
- multiple downsampling levels of the layer outputs (2x and 4x max-pooling are used together)
- feature detection at multiple scales
- capturing positive and negative phases of filters (as recommended in [24])

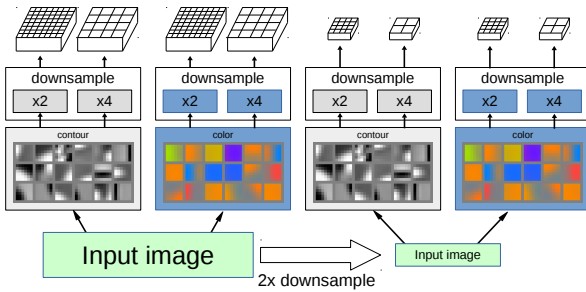

**Fig. 5.** Multiscale multimodal feature extraction layer

One distinctive trait of our approach to feature detection is high redundancy of the description produced. Neural architectures with dense connectivity try to decrease channels count as much as possible due to quadratic dependency between channels count and coefficients count. However, in our architecture coefficients count scales *linearly* with network width due to the extreme sparsity of subsequent layers ($O(1)$ connections per neuron). Thus, having a wide and redundant feature extraction layer puts much less stress on the computational budget.

## 7    Training Sparsely Connected Layers

This section discusses the core contribution of our work — the constructive training of sparsely connected strong neurons.

### 7.1    The Constructive Training Algorithm

Training networks composed of nonconvex and nonsmooth elements is difficult. It is especially difficult with $min$-based activation functions because $min$ function is extremely nonconvex and makes training prone to stalling in bad local extrema.

    Suppose, however, that *somehow* you can train just one such element to fit some target function of your choice. How can it help you train a network? The answer is to build your model incrementally, training new elements to fit the current residual and adding them one by one (see Figure 6). Every time you add a neuron to the layer you have to retrain the classifier to obtain new residuals. By feeding new neurons with outputs of the previous ones we can generate multilayer network.

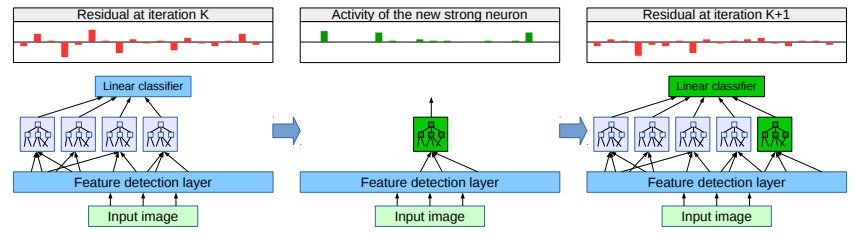

**Fig. 6.** Incremental training procedure

    Similar approaches were investigated many times ([11], [6]). The latter (Cascade-Correlation architecture) is the one which inspired our own research.

### 7.2    Training Strong Neurons

In the subsection above, we reduced the problem of training sparse multilayer networks to training just one neuron with sparse connections:

$$\min_w \sum_i \left(N(w, X_i) - y_i\right)^2 \quad s.t. \ sparsity \ constraints \quad (3)$$

    where $w$ is a weight vector, $X_i$ is an $i$-th row of the input activities matrix $X$ (activities of the bottom layer at $i$-th image), $N(w, x)$ is a neuron output and $y_i$ is a target to fit (in our case, the current residual).

For a three-input strong neuron with binary weights, the formulation above becomes:

$$\min_{w} \sum_{i} \left[ \min \left( \max_{j}(w_{0,j} \cdot X_{i,j}), \ \max_{j}(w_{1,j} \cdot X_{i,j}), \ \max_{j}(w_{2,j} \cdot X_{i,j}) \right) - y_i \right]^2 \tag{4}$$

$$w_{0,j}, w_{1,j}, w_{2,j} \in \{0, 1\}$$
$$\|w_0\|_0 \leq k \ , \quad \|w_1\|_0 \leq k \ , \quad \|w_2\|_0 \leq k$$

The problem (4) is a discrete optimization problem. There is likely no other way to solve it except for a brute-force search (no obvious reduction to mixed integer LP/QP). However, we do not need an exact solution — having a good one is sufficient. Our insight is that there is a simple heuristic that can generate good strong neurons in a reasonable time.

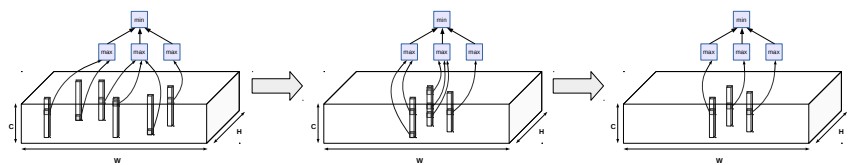

**Fig. 7.** Progressive simplification of the optimization problem 4

The original discrete optimization problem has no constraints except for sparsity. A *max*-element can gather information from any element of the input tensor (see figure 7, left). As a result, we have to evaluate prohibitively large amount of possible connection structures. For instance, for 15 unit-weight connections to elements with a 32x32x20 input tensor we have roughly $10^{58}$ possible geometries.

It is possible to significantly reduce the configuration count by adding some additional restrictions on the inter-layer connections. For example, we may impose two additional constraints: (a) require that *max*-elements are spatially local (i.e., each element gathers inputs from just one location $(x, y)$ of the input tensor), and (b) require that *max*-elements feeding data into the same *min*-element are located close to each other.

Alternatively — for 1x1xD input tensors with no spatial component — these restrictions can be reformulated as follows: (a) require that *max*-elements are correlationally local (i.e., each element gathers inputs from strongly correlated channels), and (b) require that *max*-elements feeding data into the same *min*-element are correlated strongly enough.

Having such constraints on the connections of the strong neuron significantly reduces the number of configurations that must be evaluated to solve the problem (figure 7, center). In our toy example, the configuration count is reduced from $10^{58}$ to just $10^{18}$.

We can achieve a further reduction in search complexity through a two-step search procedure: (1) evaluate all possible "seed detectors" — strong neurons with single-input $max$-elements (AND without OR), and (2) expand the best seed found — sequentially add connections to its $max$-elements.

As a result of this improvement (see figure 7, right), the search complexity for our 32x32x20 example is reduced from $10^{18}$ to $10^{9}$ neural configurations. However, it is still too costly — each of these configurations requires a full pass over the entire dataset in order to evaluate the neuron's performance.

Further improvements can be achieved by assuming the following:

– Good $f_3 = \min(A, B, C)$ can be found by extending good $f_2 = \min(A, B)$ with the best-suited $C$
– Good $f_2 = \min(A, B)$ can be found by extending good $f_1 = A$ with the best-suited $B$
– Good $f_1 = A$ can be found by simply evaluating all possible single-input seed detectors

This improvement finally makes original discrete optimization problem computationally tractable. For example, the complexity of our toy example is reduced to just 20000 combinations (compare this with the initial $10^{58}$ estimate).

*Algorithm outline.*

1. Setup the initial model (empty with zero output) and a vector of its residuals over the entire dataset. Select a neuron pool size $P$ (a few hundreds works in most cases).
2. Competition phase: generate seed detectors and select the winner from the combined pool:
   – Select a set of $P$ promising input features, "gen-1 seeds," $f_1 = A$. Some form of quick and dirty feature selection is usually enough.
   – Produce $P$ gen-2 seeds by extending gen-1 seeds $f_1 = A$ with such $B$ that $f_2 = \min(A, B)$ produces the best linear fit to the current residual. Only the spatial/correlational neighborhood of $f_1$ is evaluated.
   – Produce $P$ gen-3 seeds by extending gen-2 seeds $f_2 = \min(A, B)$ with such $C$ that $f_3 = \min(A, B, C)$ produces the best linear fit to the current residual. Only the spatial/correlational neighborhood of $f_1$ is evaluated.
3. Generalization phase. Having determined a winning seed detector, sequentially extend its inputs with new $max$-connections:
   – $f = \min(A, B, ...)$
   – $A \rightarrow \max(A)$
   – $\max(A) \rightarrow \max(A, A_2)$
   – $\max(A, A_2) \rightarrow \max(A, A_2, A_3)$ and so on
   Extending is performed in such a way that the extended detector fits the residual better than its previous version. Only the spatial/correlational neighborhood of $A$ is investigated. The procedure stops after the maximum number of connections is formed (good value — 5 connections per $max$-element) or when there is no connection that can improve the fit.

4. Add a detector to the model, and update the classifier and residual vector. Stop after the user-specified amount of detectors is formed. Go to 2 otherwise.

The algorithm above is a batch algorithm — it requires us to keep an entire dataset in memory and make a full pass over it in order to generate new strong neurons. The reason for this is that the algorithm has no way of correcting the neuron structure once it has been added to the model — so, if you train a suboptimal neuron using a subsample of the entire training set, you will be unable to improve it later.

This property raises an old question of the balance between network stability and its plasticity. Networks trained with SGD have high plasticity but zero stability. Plasticity allows us to use SGD — an algorithm that makes only marginal improvements in the network being trained — because these small decrements in the loss function will accumulate over time. At the same time, it impedes cheap nondestructive retraining — once an image is removed from the training set, it is quickly forgotten.

In contrast, our algorithm has zero plasticity — it will not improve the neurons it generated previously — but perfect stability. The drawback of such an approach is that it is necessary to use an entire training set to generate just one strong neuron, and this job has to be done in the best way possible. The upside is that the network never forgets what it learned before. If your task has changed a bit, you can restart training and add a few new neurons without damaging previously learned ones.

## 8    Training Shallow Classifier Layer

Our proposed strong neurons have a rigid piecewise linear output with a fixed slope, but in order to separate image classes one often needs nonlinearities with steep slopes in some places and flat spots in other parts of the feature space. Hence, a separate classifier layer is needed at the top of the network.

*One important point to note is that the shallow classifier layer is the only place in our model where significant adversarial instability is introduced.* The initial feature detection layer is a single layer of convolutions with bounded coefficients, and thus it has limited adversarial perturbation growth. The sparsely connected layers of strong neurons do not amplify adversarial perturbations.

As a result, any adversary targeting our model will actually target its last layer. In effect, this means that we reduced the problem of building a robust deep classifier to one of building a robust *shalow* classifier.

Our experimental results show that due to the stability of the bottom layers and computational power of strong neurons a simple logistic model (linear summator + logistic function) on top of the network performs well enough in terms of accuracy and adversarial stability.

# 9   Experimental Results

## 9.1   Datasets, Software and Network Architectures

We tested Contour Engine on two popular computer vision benchmarks: German Traffic Sign Recognition Benchmark [26] and Street View House Numbers dataset [22].

Our neural architecture is quite nonstandard, and no present framework can train such models. Thus, we had to write the training and inference code in C++ from scratch. The code — an experimental GPL-licensed machine learning framework with several examples — can be downloaded from the following link: **ANONYMIZED://URL/**

We evaluated three versions of the same architecture, listed in Table 1: Contour Engine Micro (ultralightweight version), Contour Engine and Contour Engine 800 (ultrawide contour-only version without color block, intended for SVHN dataset).

We should note that all three architectures have a non-convolutional sparse part (one that is composed of strong neurons). In theory, the constructive training algorithm described in section 7 can be applied to convolutional connection structures (it just needs a bit more coding). However, in this first publication we decided to limit ourselves to the simplest architecture, which achieves interesting enough results.

**Table 1.** Three Contour Engine versions and their parameters

| Name | contour features | color features | multiscale processing | KFLOP | width per class | KFLOP per class |
|---|---|---|---|---|---|---|
| | Unsupervised feature extractor | | | | Classifier columns | |
| CE-micro | 10x4x4 | 10x4x4 | 16x16 | 162 | 50 | $\approx 1$ |
| CE-basic | 50x6x6 | 10x4x4 | 32x32, 16x16 | 5075 | 200 | $\approx 7$ |
| CE-800 | 50x6x6 | — | 32x32, 16x16 | 4590 | 800 | $\approx 28$ |

## 9.2   Results

Table 2 examines Contour Engine performance in two categories: (a) ultralightweight networks with sub-megaflop inference cost, and (b) networks with higher computational budget (and higher accuracy requirements).

Reference results for other architectures are cited from [7] (EffNet, ShuffleNet, MobileNet), [3] (Targeted Kernel Networks: TSTN, STN), [12] (pruned VGG) and [23],[3] (Capsule Networks).

The *Contour Engine Micro* clearly outperforms other lightweight networks (EffNet, ShuffleNet, MobileNet) by a large margin. It provides superior accuracy while working under smallest computational budget.

Medium-sized Contour Engine also shows impressive results on GTSRB dataset. We have to note, however, that its test set error at SVHN is somewhat larger than that of competing approaches. Visual investigation of misclassified examples shows that it can be attributed to the fact that the SVHN dataset contains large proportion of misaligned (badly centered) images — convolutional models better generalize to such images than the nonconvolutional network which we test here.

**Table 2.** FLOPs vs Accuracy at GTSRB and SVHN datasets. Results are ordered by FLOP count (ascending)

| Network | FLOP | Error | Network | FLOP | Error |
|---|---|---|---|---|---|
| GTSRB, lightweight | | | SVHN, lightweight | | |
| **CE-micro** | **0.2M** | **6.7%** | **CE-micro** | **0.2M** | **10.1%** |
| EffNet small | 0.3M | 8.2% | EffNet small | 0.5M | 11.5% |
| ShuffleNet | 0.5M | 11.0% | ShuffleNet | 0.7M | 17.3% |
| MobileNet v1 sml | 0.5M | 11.9% | MobileNet v1 sml | 0.8M | 14.4% |
| MobileNet v2 sml | 0.7M | 9.3% | MobileNet v2 med | 1.2M | 13.3% |
| MobileNet v2 med | 1.1M | 7.2% | MobileNet v2 big | 2.1M | 12.8% |
| GTSRB | | | SVHN | | |
| **CE-basic** | **5.3M** | **1.6%** | **CE-800** | **4.8M** | **4.8%** |
| TSTN | 55.7M | 1.5% | CapsNet | 41.3M | 4.3% |
| STN | 145.0M | 1.5% | VGG-16 pruned | 210.0M | 3.9% |
| VGG-16 pruned | 522.9M | 1.2% | | | |

**Table 3.** Adversarial stability at SVHN dataset. Results are ordered by attack success rate (ASR).

| $\epsilon = 0.01$ | | $\epsilon = 0.02$ | | $\epsilon = 0.03$ | |
|---|---|---|---|---|---|
| Defense | ASR | Defense | ASR | Defense | ASR |
| no protection | 83.4% | no protection | 96.4% | no protection | 98.6% |
| Wong | 33.7% | Wong | 58.9% | IAT | 52.8% |
| **CE-800** | **18.9%** | ATDA | 46.8% | **CE-800** | **46.4%** |
| | | **CE-800** | **29.7%** | | |

Finally, we tested the adversarial stability of the *Contour Engine 800* network trained on the SVHN dataset (with clean test set error equal to 4.8%). We used a powerful PGD attack (iterated FGSM with 20 iterations and backtracking line search) with the perturbation $L_\infty$-norm bounded by $\epsilon = 0.01$, $\epsilon = 0.02$ and $\epsilon = 0.03$.

Table 3 compares the attack success rate with reference values from three independent works ([15] for Wong defense, [25] for Adversarial Training with Domain Adaptation, [16] for Interpolated Adversarial Training).

It can be seen that an unprotected network can be successfully attacked in 83% cases with a perturbation as small as 0.01. Different kinds of adversarial protection (when used on traditional summator-based networks) significantly reduce the attack success rate. However, in all cases Contour Engine outperforms these results without any special counter-adversarial measures.

## 10    Summary

In this work, we have proposed a novel model of the artificial neuron — the strong neuron — which can separate classes with decision boundaries more complex than hyperplanes and which is resistant to adversarial perturbations of its inputs. We proved that our proposal is a fundamental and well-motivated change and that the constituent elements of our strong neuron, $min/max$ units, are the only robust implementations of the AND/OR logic. We also proposed a novel training algorithm that can generate sparse networks with $O(1)$ connections per strong neuron, a result that far surpasses any present advances in neural network sparsification.

State-of-the-art efficiency (inference cost) is achieved on GTSRB and SVHN benchmarks. We also achieved state-of-the-art results in terms of stability against adversarial attacks on SVHN — without any kind of adversarial training — which surpassed much more sophisticated defenses.

One more interesting result is related to our decision to separate unsupervised feature detection and supervised classification. We found that Contour Engine spends most of the inference time in the unsupervised preprocessor — less than 10.000 FLOP per class is used by the supervised part of the network (one which is composed of strong neurons). This result suggests that contour recognition is much simpler than was previously thought!

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
