# OpenReview forum: "On sparse connectivity, adversarial robustness, and a novel model of the artificial neuron"
_thecvf.com/ECCV/2020/Workshop/VIPriors — VIPriors Oral_

### Official Review · AnonReviewer1 · 2020-07-21
**A novel approach to counter recognition**

**Confidence:** 3
**Rating:** 7

**Review:**

[Summary] In 2-3 sentences, describe the key ideas, experiments, and their significance.

The authors propose a new type of neuron designed for contour recognition. They detail an extensive algorithm for training these neurons without back-propagation. They show their method can outperform convolutional methods in low FLOWs regime and is more robust against adversarial attacks.

[Strengths] What are the strengths of the paper? Clearly explain why these aspects of the paper are valuable.

Creative approach to a hard problem (replacing the convolutional neuron); builds on related work where possible; solid experiments.

[Weaknesses] What are the weaknesses of the paper? Clearly explain why these aspects of the paper are weak.

Many heuristics are needed to optimize the strong neuron, while the effect of the heuristics are not analyzed or explored. In the same vein, an ablation study, including the effects of the unsupervised backbone, would have helped to make the work more solid.

[Overall rating] Paper rating: Accept

[Detailed comments] Additional comments regarding the paper (e.g. typos or other possible improvements you would like to see for the camera-ready version of the paper, if any.)

- Lines 156-157, 189-190 are unclear to me
- Section 7.1 addresses the reader as "you". Use of more formal "one" would be advised.
- Change brackets to separate sentences (e.g. lines 370-371)
- Minor typos: lines 490 "shalow"

---

### Official Review · AnonReviewer2 · 2020-07-22
**Interesting and novel idea**

**Confidence:** 3
**Rating:** 8

**Review:**

#### 1. [Summary] In 2-3 sentences, describe the key ideas, experiments, and their significance.
The paper proposes a novel, computationally efficient artificial "strong neuron" for sparse neural networks that combines low-level features through AND and OR operations and a corresponding training strategy. The resulting networks are evaluated on the GTSRB (German traffic sign) and SVHN datasets and show competitive results in both classification error and adversarial stability.

#### 2. [Strengths] What are the strengths of the paper? Clearly explain why these aspects of the paper are valuable.
* The proposed "strong neurons" are definitely a novel and interesting idea and the motivation is clearly explained through Figure 1.
* The method seems effective on the evaluated datasets.
* The paper is generally well written and easy to follow.
* Code is made available which will hopefully spark interest for research into alternatives to traditional CNNs.

#### 3. [Weaknesses] What are the weaknesses of the paper? Clearly explain why these aspects of the paper are weak.
* The optimization method is based on a lot of heuristics to simplify the otherwise intractable brute force approach. Although the optimization method seems effective (based on the performance), it is hard to evaluate the possible negative effects of these simplifications on the model performance.
* The training strategy not allowing to use mini-batches seems like a major drawback for training on large-scale or high resolution datasets like ImageNet or CityScapes.

#### 4. [Overall rating] Paper rating
* 8. Top 50% of accepted papers, clear accept

#### 5. [Justification of rating] Please explain how the strengths and weaknesses aforementioned were weighed in for the rating.
The authors have proposed a novel and interesting alternative to traditional CNNs and have shown its effectiveness.

#### 6. [Detailed comments] Additional comments regarding the paper (e.g. typos or other possible improvements you would like to see for the camera-ready version of the paper, if any.)
* Combine multiple references into same brackets, i.e. ([1], [2], [3]) should be [1,2,3].
* What do $k$ and $i$ represent in equation (1) and line 125/126?
* The paper would be easier to read if a conclusion would be included in the image captions (i.e. what point is the image trying to make).

---

### Decision · Program_Chairs · 2020-07-29

**Decision:**

Accept (Oral)

**Comment:**

It is our pleasure to inform you that your paper has been accepted to the oral track of the 1st Visual Inductive Priors for Data-Efficient Deep Learning Workshop.

Please note the following deadlines:
* August 11, 2020 - workshop material, including:
 * paper in PDF format;
 * pre-recorded video presentation;
 * slides of the presentation in PDF.
* September 15, 2020 - camera-ready paper

The reviews can be found on OpenReview. Please take these comments and suggestions into account when preparing the camera-ready version of your paper, which is due September 15, 2020. The camera-ready paper should be uploaded to OpenReview.

As part of the workshop, each paper for oral presentation must submit a pre-recorded 5 minute talk before August 11, 2020. You will receive more information on how to upload the material shortly. The requirements for the video are:
* Duration: maximum 5 minutes
* MP4 format
* File size max. 100 MB
* Has an inset with a video of the speaker
* 16:9 aspect ratio (strongly preferred)
* 1920x1080 resolution (strongly preferred, at least 720 height)

Our suggested software for pre-recording your presentation is Zoom. For more information, please refer to the following guides:
How to record with Zoom Guide: http://homepages.inf.ed.ac.uk/rbf/ECCV2020HowtoRecordusingZoom.pdf
How to Record with Zoom tutorial: https://www.youtube.com/watch?v=CR199W7HdC0
Please ensure that at least one of the authors of the paper is available to attend the workshop during the allotted times. Note that the workshop will take place in two sessions spread across time zones (details are to follow). We will send instructions on how to connect to the workshop as soon as possible. The schedule for all talks and papers will be posted soon at the workshop website: https://vipriors.github.io.

We look forward to seeing you at the workshop!